# Study and Application of Asymmetrical Disk Tools for Hard Rock Mining

**Krzysztof Kotwica** [1],*, **Grzegorz Stopka** [1] **and Dariusz Prostański** [2]

1   Faculty of Mechanical Engineering and Robotics, AGH University of Science and Technology, Mickiewicza Ave. 30, 30-015 Kraków, Poland; stopka@agh.edu.pl
2   KOMAG Institute of Mining Technology, Gliwice, Pszczyńska 37, 44-101 Gliwice, Poland; dprostanski@komag.eu
*   Correspondence: kotwica@agh.edu.pl; Tel.: +48-607467068

**Abstract:** In this article, the method of hard rock mining by undercutting or back incision using asymmetrical disk tools as an alternative to the milling method with the use of cutting tools was described. The results of modelling the penetration of a single asymmetric disk tool edge into an artificial and natural rock sample were presented and compared with empirical laboratory tests. The effect of stand tests for mining artificial rock samples using asymmetrical disk tools mounted on the rotating plate was presented. The tests were carried out on two unique test stands. The solution of an innovative mining head with a complex motion trajectory, using the asymmetrical disk tools, was presented. The innovative mining head especially developed at the AGH Kraków Department of Machinery Engineering and Transport for roadheaders, was presented. The results of preliminary field tests with the use of this mining head were also described.

**Keywords:** asymmetrical disk tool; hard rock mining; modelling; mining head; tool wear; efficiency



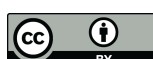

## 1. Introduction

Gallery and tunnel driving and mineral resources excavation are currently often performed under severe conditions. These relate to the significant depth of the deposits and more challenging geological conditions. The mined rock has a considerable compressive strength value, sometimes over 200 MPa, and may include in many cases inclusions of abrasive minerals such as, for example, spherosiderites. The costs incurred to uncover deposits and perform preparatory work have a significant effect on mining's economic efficiency. That is why the mechanical complexes to facilitate driving preparatory and exploitation workings at advance rates high enough to reduce mining costs, are used. One of the main tasks when selecting these complexes is choosing an appropriate method of rock mining, ensuring the durability of the mining tools and low energy consumption of the mining process [1–5].

The separation of output chip from from the solid rock, i.e., the mining process, is the first operation in a sequence of technological processes involved in gallery driving or useful-mineral mining. As such, it is a core mining operation. This process is used not only in mining. Such procedures are also standard in construction works involving excavations, tunnels and digging foundations, etc. [2–7].

A definition of the mining as a process in which rock pieces are separated from the solid does not fully reflect the nature of the process and may lead one to consider the mining as being equivalent to the physical process of rock breakage, which is essentially an energetic process.

In addition to energy transformation, mining must involve information processing to control the primary rock structure's disintegration such that the objective can be achieved. The additional task is to extract rock in a specific geometric range (a gallery profile shape) and specific cross-sectional parameters.

Rock mining's primary purpose is to separate from the rock massive the largest possible rock pieces using as little energy as possible. Rock mining is generally measured using the specific energy $E_w$ [1,5–7]:

$$E_w = \frac{E_u}{V} \tag{1}$$

where $E_u$—energy supplied to excavate the rock (J) and $V$—volume of the excavated rock, ($m^3$).

Thus, the specific energy expressed by Equation (1) is the energy required to excavate a unit of a rock's volume, the lower the specific energy, the more efficient the process. During rock mining, energy is used to break the rock's structure by creating cracks and craters. Hence, it seems reasonable to assume that the supplied energy $E$ will be proportional to the newly created surface (~$d^2$, where $d$ is the linear dimension of the newly formed rock grain) in an efficient process. Thus:

$$E_w \sim \frac{1}{d} \tag{2}$$

Hence, the specific mining energy is reversely proportional to the dimensions of the removed rock piece. It means that as the grain dimensions increase, the energy required to excavate a unit of the rock's volume decreases. It is commonly known as Rittinger's law.

In mechanical complexes used for gallery and tunnel driving and mineral resources excavation, the most common methods of mechanical rock mining, such as cutting and milling are applied. The cutting tools such as radial and rotary-tangential picks are mainly mounted on mining heads of roadheaders and shearers [2,8].

The mining of rocks with highly unfavourable parameters, using cutting tools, has been an issue for the mining industry. This is connected with highly compact and an enormous uniaxial compressive strength of the excavated rock and an almost homogeneous structure of the rock and hard and abrasive minerals and inclusions in the stone. In the case of cutting tools, it can generate three threats during mining (Figure 1). Those are sparking (Figure 1a), dustiness (Figure 1b) and increased pick-edge wear (Figure 1c) [2,9–14].

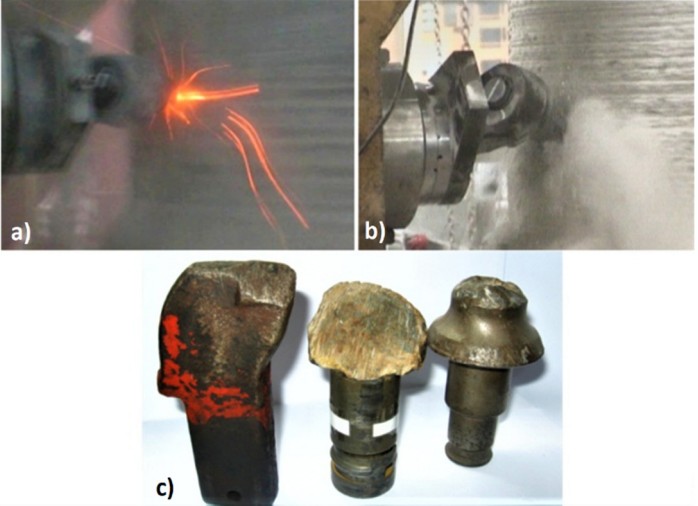

**Figure 1.** View of the threats generated during cutting of a rock sample with cutting tools: (**a**) frictional sparking, (**b**) dust generation, (**c**) pick edge wear [2].

For elimination or decreasing of these threats, in mechanical mining are used other non-conventional, mechanical mining methods using disk tools [2,15–17]. Those are:

- rock mining based on static crumpling using symmetrical and asymmetrical disk tools.
- undercutting or back incision with asymmetrical disk tools.

## 2. Purpose, Course and Methodology of the Research Activities

The main goal of this study was to determine the possibility of effective mechanical mining of hard rock with the use of asymmetrical mini-disk tools, obtaining large granulation of the output and lower energy consumption compared to mining with the use of symmetrical disk tools. In the first part, the principle of the mentioned above mechanical mining methods using disk tools was described. Also, the benefits and faults of these methods were presented.

The results of model tests of the compact rock mining process with asymmetrical disk tools developed and carried out at the AGH Kraków Department of Machinery Engineering and Transport are presented. The Discrete element method (DEM) with the LS-Dyna computer package (Livermore Software Technology Corporation, Livermore, CA 94551, USA) was used to carry out these studies. The goal of the model tests was to identify the mining resistance with a single asymmetrical mini-disk tool, especially the mining process energy parameters. During simulation tests, normal (pressing) force (Pd) and side force (Pb) values acting on the asymmetrical mini-disk tool were monitored.

Based on the obtained results, the disk tools' geometrical parameters and the parameters of the mining process were selected, which were then used in the laboratory verification tests.

Successive attempts of pressing the asymmetric mini-disk tools into the concrete sample, mining with plate equipped with asymmetrical mini-disk tools with straight-motion trajectory and mining with a complex-motion trajectory milling plate, equipped with asymmetrical mini-disk tools were carried out. During the tests, the influence of the mining spacing, mining depth, the solution of mining method and the geometry of disk tools on the cutting resistance and granulation of the excavated material was examined.

The results of model studies and laboratory tests enabled the selection of parameters and the execution of the innovative solution of mining head with mini-disk tools of complex motion trajectory. With the use of a new mining head mounted on a roadheader, field tests of the mining process effectiveness were carried out on a large-size concrete block. The most advantageous parameters of the mining process (the direction and number of rotations of the mining head as well as the mining method) were determined to obtain the highest output grain size distribution at the lowest energy demand.

## 3. Mechanical Mining Using Disk Tools

For gallery driving under challenging conditions, the mechanical method currently uses symmetrical or asymmetrical disk tools. The technique of rock mining based on static crumpling using symmetrical and asymmetrical disk tools is shown in Figures 2 and 3. In this method, the disk's edge, with a V-like, symmetrical or asymmetrical shape of its cross-section in the plane normal to its edge, is pushed into the rock with Pd's pressure force perpendicular to its surface [2,15,18].

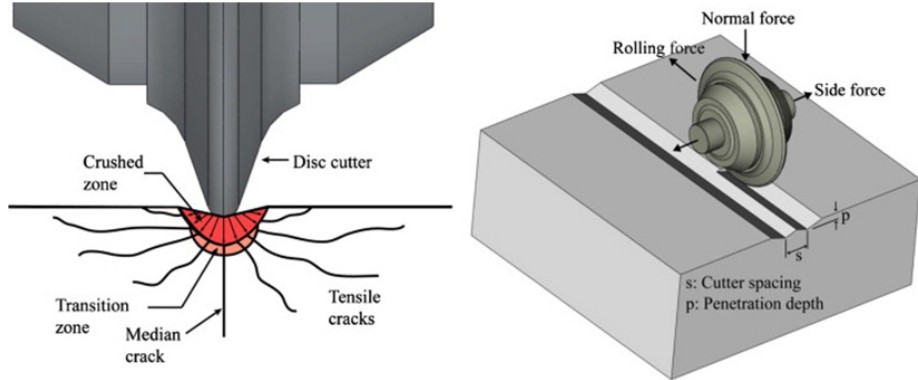

**Figure 2.** The method of rock mining based on static crumpling using symmetrical disk tools [18].

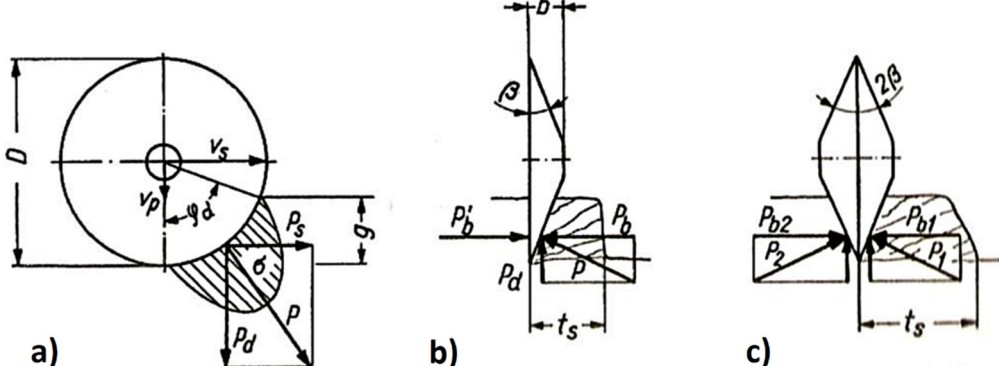

**Figure 3.** The method of rock mining based on static crumpling: (**a**) general scheme, (**b**) the way of mining using asymmetrical disk tool, (**c**) the way of mining using symmetrical disk tool [15].

As a result of this force, the rock's compressive strength is locally exceeded, with the disk driving into it to the depth g. Besides, the tangential force Ps is applied to the disk handle, causing the tool to move along the rock's surface. Due to tangential Ps and pressure Pd forces, the disk puts the rock under pressure coming from the resultant force P along with a circumference corresponding to the angle $\varphi_d$ (Figure 3). The disk is mounted in a rotary holder, enabling it to rotate on its axis with the angular speed $\omega$ [15].

The fundamental advantage of this method is that it radically reduces friction forces' role by using the tool's rotary movement. Moreover, due to rotary motion, each section of the disc's edge remains in contact with the rock for a short period, during which it does the work required to achieve the target notch depth. It results in smaller energy losses, and better heat dissipation conditions on the disc's edge, reducing the side effects of mining, such as sparking due to local increases in temperature, or dust generation, and extending the service life of the tool. This method can be used for the mining of rock with compressive strength over 200 MPa [15,18].

The symmetrical disk tool diameter can reach up to 500 mm and its lifetime allows it to generate up to several kilometers of roadways or tunnels without visible wear. By comparison, a cutting tool's service life is typically no more than several cubic meters of output per one piece [19–21]. It also allows achieving the large runout of the drilled gallery or tunnel—up to 70 m per day.

The disadvantage of mining by static crushing is ensuring a high-pressure force exerted by the tool. The value of the pressure force per disk tool can be as high as 300 kN. For the whole mining head, it can generate the total pressure force value up to 25,000 kN. This forces huge dimensions (up to 450 m of length), weight (up to 3500 Mg) and cost of the machine called TBM. Therefore, such a technology is only profitable for extended excavations, over a few kilometers [2,18].

But the asymmetrical disk tools have been used on mining tools of longwall shearers to increase the output of a large size grade. Conducted industrial tests demonstrated the usefulness of such equipment for obtaining higher graining, but increased dynamics of mining with the units was why they are not commonly used nowadays [16,22,23].

Asymmetrical disk tools are applied in mechanical mining, not only as crushing devices but also as chipping ones. Mining by undercutting or back incision with a disk tool uses typically for rocks much smaller resistance against stretching than uniaxial pressing. The principle of the back incision technique is mining a rock by cutting it off towards free space [1,2,17].

Figure 4 presents a diagram of mining with the method of the back incision. Application of disk tools in that way lowers energy consumption and pressure force which allows constructing a mining machine of respectively lower energy parameters, lower requirement concerning stability than in case of classical TBM's machines operating perpendicularly towards the surface of the mined body [17].

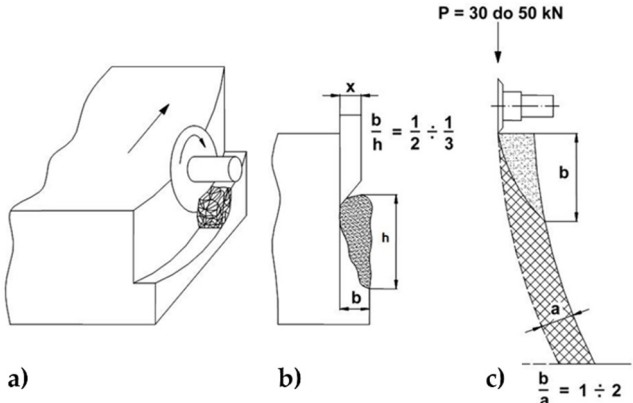

**Figure 4.** Diagram of the back incision mining method principle: (**a**) general scheme, (**b**) parameters of the output piece chipped from the solid rock, (**c**) parameters of the mining process [17].

However, in this method, there occur strongly changeable side forces on disk tools edges. It causes difficulties with proper transfer of reactions on disk tools chucks and their bearing.

The machine's prototype used an undercutting method, developed by the Wirth Company (Erkelenz, Germany), and showed the full usability of the suggested solution. The machine equipped with four independently controlled arms with asymmetrical disk tools 450 mm in diameter was applied for drilling excavation of dimensions 4 m × 4 m in relatively hard mineable rocks. The method of back incision used in the presented machine showed full usability of the suggested solution. A very high granulation of the output grains was obtained (even over 200 mm), with a unit mining efficiency of 3 $m^3$/h per disk. The only drawback is a very complicated method of steering individual arms and very high reaction forces [17].

That is why the idea of the back incision technique was taken into consideration at the AGH University of Science and Technology Krakow Department of Machinery Engineering and Transport, to elaborate new method of rock mining and innovative construction of a mining unit equipped with mini asymmetrical disk tools. The course of implemented works is presented below.

### 4. The Modelling of the Rock Mining Process with the Use of Asymmetrical Disk Tools

In the next part of the research the model tests were carried out. The goal of the model tests was to identify the mining resistance with a single asymmetrical mini-disk tool, especially the energy parameters of the mining process. Simulation research on the mining process with asymmetrical disk tools has been carried out with the use of discrete element method (DEM) using the LS-Dyna computer package,

The discrete element method (DEM) is a particle-based numerical method, which can model the mechanical behaviour of a system composed of an assembly of discrete spherical particles. During the past decade, numerical simulations of rock cutting with the use of DEM method have been carried out by many researchers. Most studies were carried out during the simulation of rock mining with the use of conical picks The DEM method has also been used to simulate rock mining by symmetrical disk tool, especially for disk tools used in full face tunnel boring machines (TBM) [22–25]. The method models parts composed of rigid spheres, that can move independently and interact with one another. DEM method can be used also to model elastic material and brittle material fracture by the bonding of the loose DEM particles. For example, in LS-Dyna computer package the properties of the bonds are adjusted by assigning normal and shear stiffness and maximum normal and shear stress for bond rupture. By adjusting the bond radius multiplier, it is possible to bond each particle to several other particles. Figure 5 illustrates a diagram of connections between particles implemented in the LS-Dyna computer package [26].

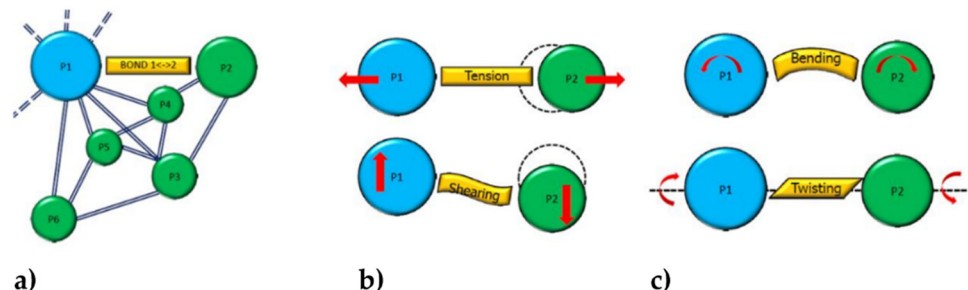

**Figure 5.** Schematic drawing of the possible bonding of particle P1 to its neighbors P2 to P6: (**a**) using intermolecular bonds, (**b**) using tensile and shear stresses, (**c**) using torsional and bending stresses [26].

In this article, the discrete element method (DEM) implemented in LS-Dyna was adopted to simulate rock cutting with asymmetrical disc tools. Numerical tests were conducted by pushing the disc into a concrete sample at a given distance from the sample edge. The independent variables in the study were the disc diameter and the cut spacing. In the simulation tests, the DEM model of the concrete sample was created with a particle radius from 1 to 2 mm. The micro and macro parameters of the DEM model were calibrated by simulating both the UCS and BTS tests.

The disk tool was treated as a rigid body with a prescribed translational motion perpendicular to the rock sample. In order to define the disk tool movement, a constant velocity of 0.1 m/s was imposed. The DEM particles were set in contact with the disc with the use of the contact node to surface contact. The static and dynamic friction coefficient was set to 0.3. During simulation tests, a displacement of the backside and right side of the sample was fixed, whereas the other sides were set to free. The material failure occurred as a result of reaching the assumed bond strength value in the normal (PBN_S = 14 MPa) or shear (PBS_S = 14 MPa) direction. These values were determined in calibration tests based on the results of UCS and BTS tests for concrete.

Simulations of rock mining by asymmetrical mini-disk tools have been performed using the model shown in Figure 6. Numerical tests were conducted by pushing the asymmetrical mini-disk tool into a rock sample at a given distance from the sample edge until the material was detached entirely. In the simulation tests, models of asymmetrical mini-disk tools with a diameter D of 150 and 160 mm and disk edge angle 40° were used [27].

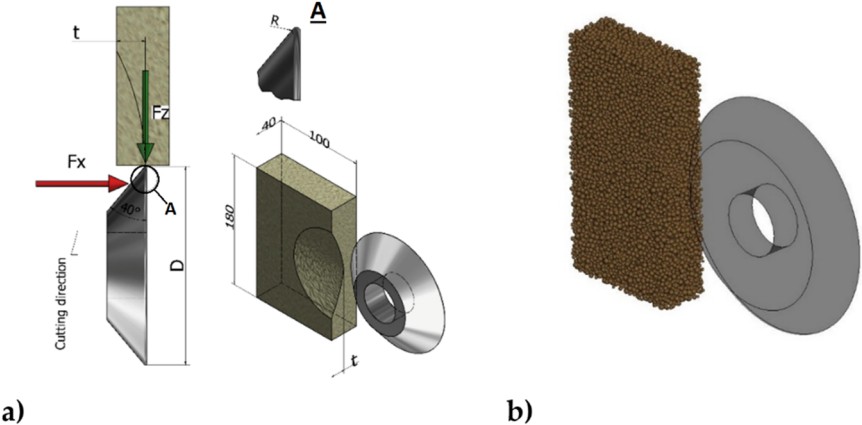

**Figure 6.** The model used for the tests: (**a**) geometrical parameters of the rock and asymmetrical mini-disk tool model, (**b**) DEM model [27].

The rock sample mining process was tested with the mining spacing of 15 and 25 mm. The research with a single asymmetrical mini-disk tool was carried out on concrete samples with uniaxial compressive strength (UCS) $R_c$ = 24 MPa and tensile strength (BTS)

$R_r$ = 2.3 MPa. In the first stage of the simulation tests, the rock sample calibration was carried out. The micro and macro parameters of DEM model were calibrated by simulating both, UCS and BTS tests. In the next step the virtual 3D model of disk tools and rock samples have been discretized [28,29].

Exemplary results of simulation tests of rock mining by asymmetrical mini-disk tool are shown in Figure 7. The figures below illustrate material failure induced by the pushing the asymmetrical mini-disk tool into a rock sample in case of disk diameter equal 160 mm. All of tests were completed by estimation of the size of the detached chip, whose dimensions (height, depth) were multiple times higher of the assumed mining spacing.

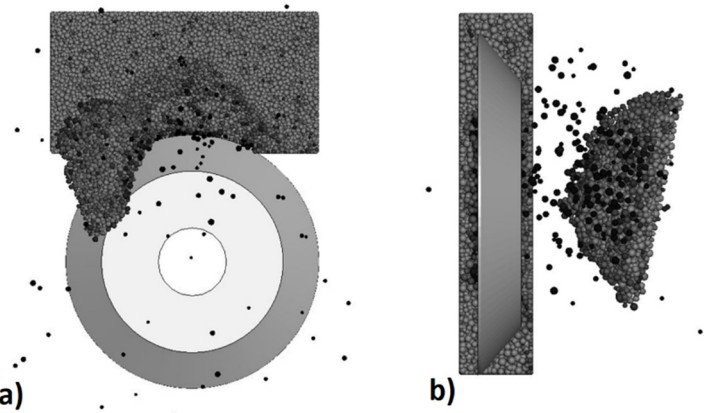

**Figure 7.** The results of the simulation test of concrete sample mining by asymmetrical mini-disk tool of diameter d = 160 mm and mining spacing t = 25 mm: (**a**) side view, (**b**) top view.

During simulation tests normal (pressing) force (Pd) and side force (Pb) values acting on the asymmetrical mini-disk tool were monitored. These parameters were the most important factors that were used to verify the adequacy of the prepared model. Figure 8 shows selected force courses for asymmetrical mini-disk tools od diameter 150 and 160 mm for mining spacing t = 15 mm and 25 mm. The recorded courses are characterized by the variability typical for the processes of rock massive destruction by a tool similar in the shape profile to the wedge. A sudden drop in the value of the forces in the graphs means the moment when the rock fragment was chipped from the block.

Based on the simulation results, it can be concluded that increasing the mining spacing from 15 mm to 25 mm causes an almost twofold increase in the pressing force. Increasing the asymmetrical mini-disk tool diameter from 150 mm to 160 mm resulted in an increase in the pressing force by about 15%. The side force Pb value is approximately about 35% of the pressing force Pd value.

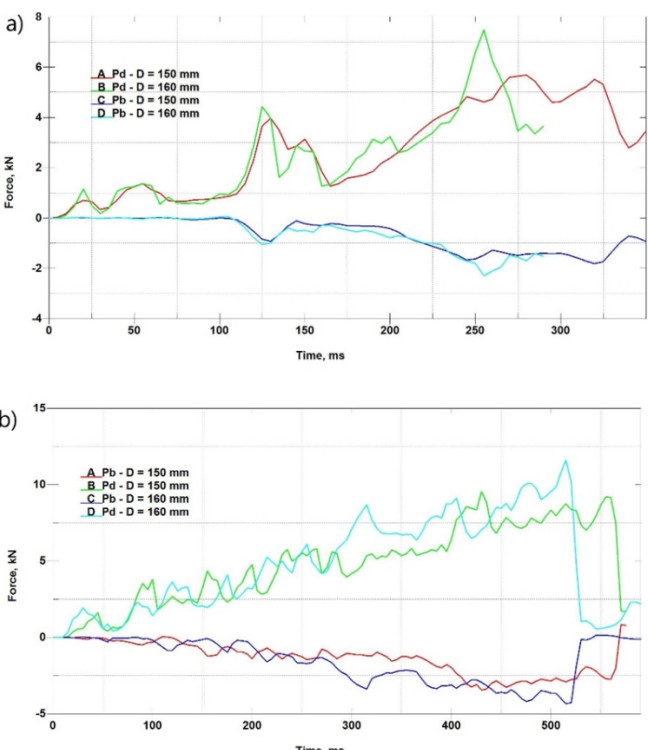

**Figure 8.** The normal and side force value courses for concrete sample mining with asymmetrical mini-disk tool: (**a**) for mini-disk tool of diameters 150 and 160 mm, (**b**) for mining spacing of 15 and 25 mm.

## 5. Verification Tests of the Rock Mining Process with the Use of Mini Asymmetrical Disk Tools at Laboratory Stands

The results of modeling the mining process with asymmetrical mini-disk tools were verified in the next part of the research during mining tests on laboratory stands owned by the Department of Engineering Machinery and Transport. Their course and the results obtained are described below.

### 5.1. Test Results on the Stand for Pressing in Asymmetrical Mini-Disk Tools

The purpose of the research was identification the impact of geometric parameters of asymmetrical mini-disk tools on the values of loads generated during pushing a disk tool into a rock sample. A detailed description of the research methodology is included in the publication [27]. Static tests consisted of pushing the disk tool into a rock sample at a given mining spacing from the sample edge until the material was detached entirely. The following geometric and process parameters were adopted as independent variables for the tests:

- the diameter and an edge angle of the asymmetrical mini-disk tool (D, $\alpha$),
- mining spacing (t),
- physical and mechanical properties of the rock sample ($R_c$).

Based on the literature analysis of the subject as well as previous author experiences in the field of industrial application of asymmetrical mini-disk tools, it has been determined that the tests will be carried out for disk tools with the diameter D = 150, 160 and 170 mm and edge angles $\alpha$ = 35°, 40°, and 45°. Figure 9 shows the view of laboratory stand for mini-disk tool pressing into rock sample and the scheme of measured parameters during mining tests. The laboratory stand used for the tests is described in more detail in Section 5.2.

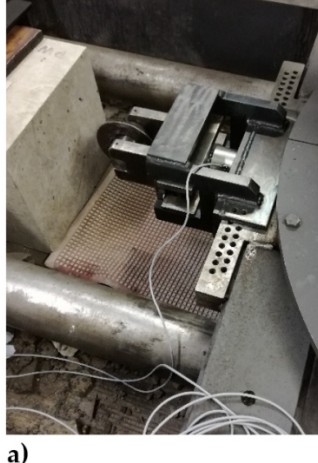
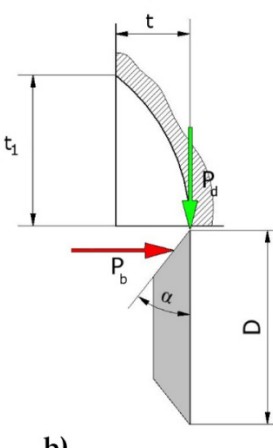

a)                                                          b)

**Figure 9.** The view of laboratory stand for testing mining process with a single mini-disk tools—(**a**) scheme of measured parameters during mining tests—(**b**).

The used test stand allowed us to attach the disc tool in the holder; the construction of this holder permitted us to change the position of the disk relatively to the sample, and thus gave the opportunity to set the right cut spacing. The system was equipped with a strain gauge force sensor that allowed measuring the pressure force in the range of 0–200 kN and a transformer displacement sensor with the measuring range of 0–300 mm.

During the conducted research, the Pd pressing force and displacement of the disk tool were measured and recorded. Research on the disk mining process was performed for concrete samples with different mechanical properties. The research with a single asymmetrical mini-disk tool was carried out on concrete samples with uniaxial compressive strength of 25 MPa and sandstone samples with uniaxial compressive strength of 79 MPa.

Each of the mining tests consisted of setting an appropriate mining spacing by the appropriate aligning and locking of the handle together with the disk tool. In the next step, the disk penetration was made close to the edge of the rectangular rock sample until the mined material was entirely deboned. The tests were carried out for three mining spacing values, namely for the parameter t equal to 15, 25 and 35 mm. After each of the tests, the extent of the destruction zone was measured. Performing another attempt required moving the holder towards the opposite, intact edge of the sample and rotating the disk tool so that the cutting was obtained by the conical surface of the disc.

In the next step, the disk penetration was made close to the edge of the rectangular rock sample until the mined material was entirely deboned. The tests were carried out for three mining spacing values, namely for the parameter t equal to 15, 25 and 35 mm. After each of the tests, the extent of the destruction zone was measured. Performing another attempt required moving the holder towards the opposite, intact edge of the sample and rotating the disk tool so that the cutting was obtained by the conical surface of the disc. Each of the trials for a given configuration of the geometric disk parameters and the determined scale pitch was repeated at least three times. Based on the observations made during the tests, there was assumed that the compression zone influence in the process of destroying the consistency of the sample was insignificant. Most tests were completed by chip separation, whose dimensions (height, depth) were multiple times higher of the assumed mining spacing. Exemplary results of static penetration of disk tools are presented in the Figure 10. The exemplary forms of destruction of sandstone samples mined with disk tools 150 mm in diameter and edge angle 40° were presented. Observations of the process of rock destruction in simulation tests, especially the shape of destruction zone, allow us to identify many similarities with the laboratory test results.

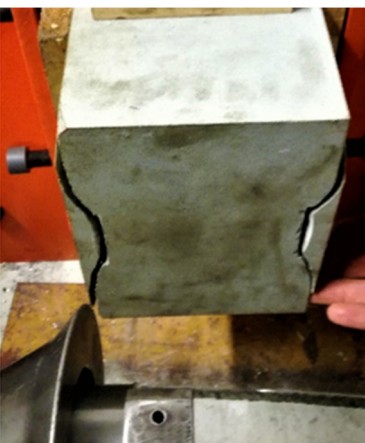

**Figure 10.** A view of the sandstone block after the miningwith disk tools 150 mm in diameter and edge angle 40°.

Figure 11 presented an example of graphs of the impact of selected geometric parameters of asymmetrical mini-disk tools and mining spacing on the value of press force. Mean values of maximum press forces were calculated based on at least three measurements. Based on the results of the research, it can be concluded that the increase in the value of individual independent variables caused an increase in the generated downforce. On the other hand, the dynamics of the changes of the press forces values varied depending on the variable adopted as the input parameter for the tests. The most significant increases in the pressing force value were observed in relation to the cutting spacing (Figure 11a,d). The change of such parameters as the diameter of the disk tool (Figure 11b) or the angle of the disk edge (Figure 11c) caused a smaller increase of the pressing force as in the case of the mining spacing. As part of the research, the influence of the compressive strength of the sample on the value of the pressing force of the disk tool was also checked. Based on the tests carried out, it can be concluded that the increase in the compressive strength of the sample caused a proportional increase in the registered value of the press force. Comparing the results obtained during simulation tests and laboratory tests, for the same mining parameters, e.g., disk diameter, mining spacing, etc., it can be stated that they are very comparable. The difference in the value of the forces does not exceed 15%.

*5.2. Test Results on the Stand for Mining with a Straight-Motion Trajectory Plate, Equipped with Asymmetrical Mini-Disk Tools*

In the next stage of laboratory tests, research with a single plate with asymmetrical disk tools of diameter 160 mm and edge angle 40° was planned. A special test stand was developed, which allowed the plate with disk tools to rotate and move in three mutually perpendicular directions. The plate's maximal diameter was defined at 450 mm, the number of disk tools at six pieces and their distribution on a diameter equal or lower than 350 mm. The 3D model of this test stand is shown in Figure 12 [30].

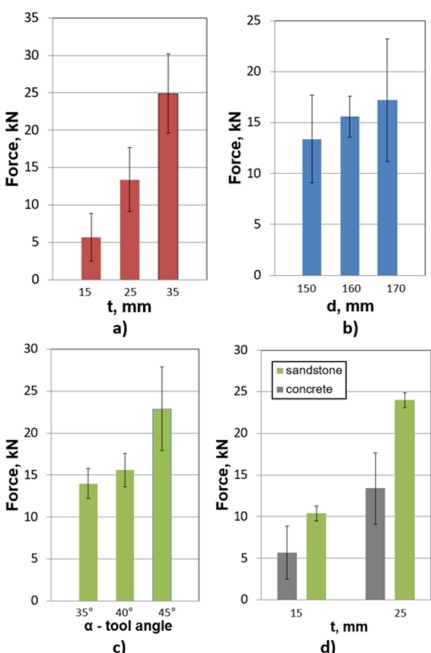

**Figure 11.** Mean values of maximum press forces for: (**a**) disk tool diameter D = 160 mm and sample compressive strength $R_c$ = 25 MPa, (**b**) mining spacing t = 25 mm, edge angle $\alpha$ = 40° and sample compressive strength $R_c$ = 25 MPa, (**c**) disk tool diameter D = 160 mm, mining spacing t = 25 mm and sample compressive strength $R_c$ = 25 MPa, (**d**) disk tool diameter D = 150 mm and sample compressive strength $R_c$ = 25 and 79 MPa.

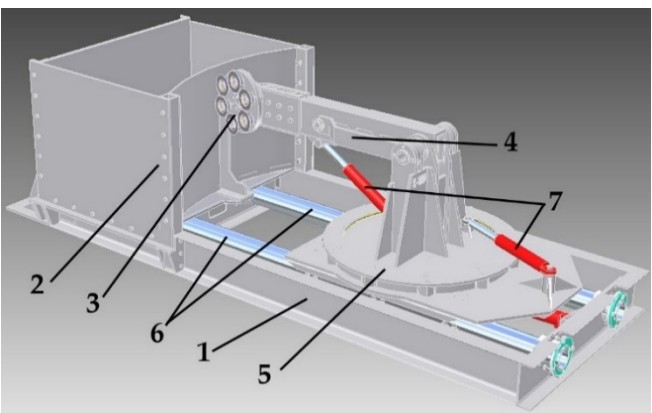

**Figure 12.** The 3D model of a lab stand for testing mining process of compact rocks with a straight-motion trajectory plate equipped with asymmetrical mini-disk tools: [30]. 1—mainframe, 2—artificial rock sample, 3—plate with disk tools, 4—jointed arm, 5—rotary plate, 6—two pipe slides, 7—two hydraulic actuators.

The main element of the test stand is the frame. In its front part, an artificial rock sample of the volume of almost 2.0 m$^3$ was installed. The plate with disk tools was propelled by a power unit comprised of the hydraulic engine of OMTS 250 type and integrated planetary gear of RR5/OMC type. The whole mining unit was mounted on the jointed arm, fixed on the rotary plate. The rotary plate was moved forward and backwards on the two pipe slides. Movement of the arm with the plate with disk tools was realised horizontally and vertically by two hydraulic actuators, raising and lowering the arm and the second one for turning the rotary plate. The stand was fed and controlled from a special hydraulic aggregate allowing free and smooth setting of the value of pressure and flow rate and change of the plate's rotation direction with disk tools. The aggregate parameters

enabled to obtain the plate's number of rotations with disk tools within the range from 0 to 65 rpm [2,30].

It was planned to perform tests comprising at least four changing parameters:

- Direction and number of the plate rotations with disk tools (overshot and undershot rotations and three ranges of rotation velocity 40, 50, and 60 1/m).
- Mining spacing from 10 mm to 40 mm.
- Cutting depth from 6 to 20 mm

The above-listed parameters were assumed to perform cuts moving the plate with disk tools horizontally and then vertically against a concrete sample [30]. A diagram of mining the rock in both planes was presented in Figures 13 and 14. Depending on mining direction (left or right) and vertical mining of the sample from the left or right edge, the disk tool attacks the rock body with its flat or inclined surface.

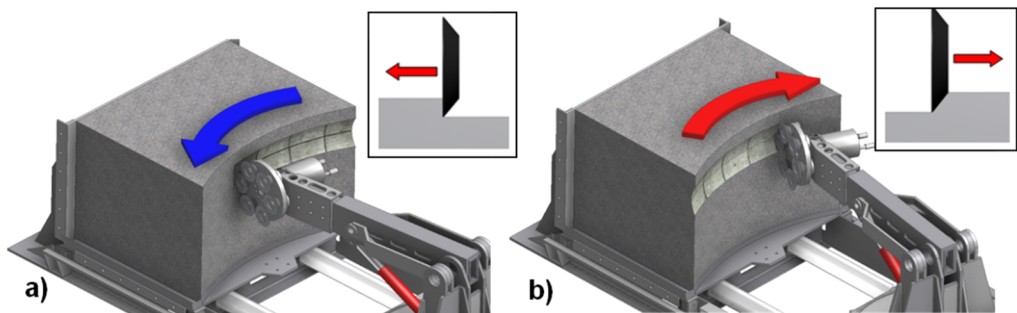

**Figure 13.** The scheme of horizontal mining of the samples: (**a**) to the left, (**b**) to the right [2,30].

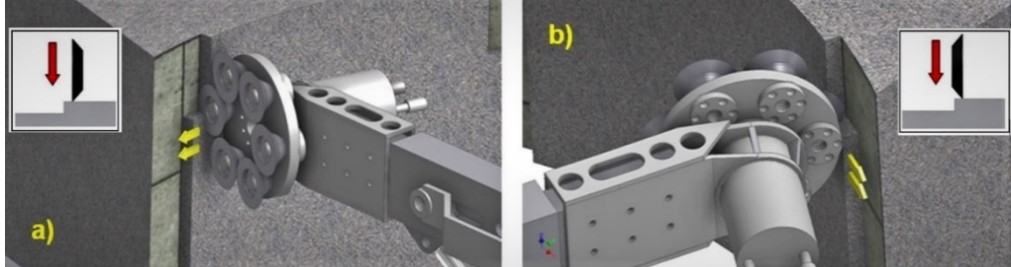

**Figure 14.** The scheme of vertical mining of the sample: (**a**) from left edge, (**b**) from right edge [2,30].

The first test was conducted on a concrete sample of uniaxial compressive strength of 26 MPa. The concrete sample was 1450 mm in width, 1300 mm in depth and 1050 mm in height. Some mining tests were repeated on natural rock samples embedded in a concrete block. Sandstone blocks featuring uniaxial compressive strength $R_c$ = 73 MPa and a granite block of $R_c$ = 253 MPa were used for the tests. Both natural rock samples were 350 mm in width, 300 mm in depth and 950 mm in height.

Mining tests in a horizontal plane were all conducted on a concrete block. The depth varied from 0 to 10 mm, whereas the plate rotation rate was 40 to 60 rpm. At the speed of 60 rpm, a significantly lower amplitude of the construction vibrations was observed, which considerably improved the disk plate's work conditions. The disk plate motion's overshot direction was applied to minimise vibrations of the construction stand [30].

For this value and the directions of rotation of the plate, the tests started, which included passes of the disk plate on the sample's full width. The cutting depths were within the range from 6 to 15 mm, implementing passes horizontally to the left and right alternately. At the plate's move to the left side of the block, a significantly lower dynamics of the mining process was observed, resulting from attacking the sample with the disc's flat surface. A view of registered courses of component forces values (pressure Pd, tangential Ps

and side Pb) during mining to the left and right at mining depth g = 10 mm was presented in Figure 15 [30].

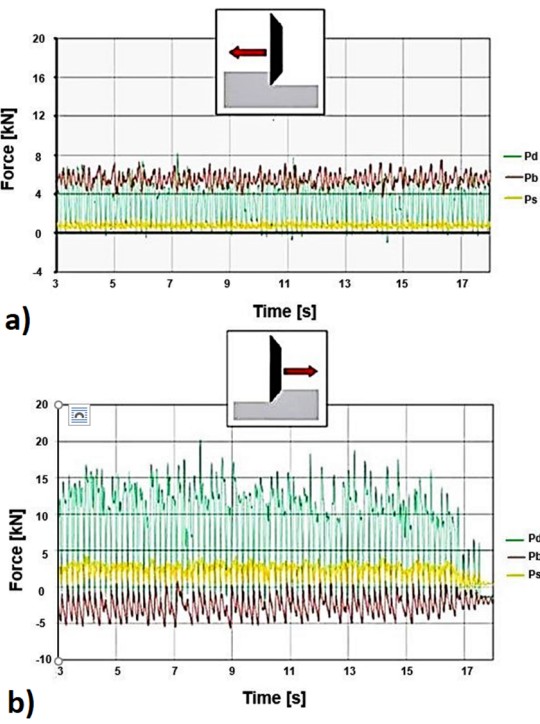

**Figure 15.** Courses of values of pressure Pd, cutting Ps and side Pb forces during mining at mining depth of g 10 m: (**a**) to the left, (**b**) to the right [30].

The main part of the study included mining tests in a vertical plane that were implemented firstly on the concrete block and then on natural sandstone and granite blocks embedded in a concrete block. Tests of vertical mining started with mining the concrete block. Firstly, the opening cut at a depth of 10 mm was performed. Then, half-open cuts were done for cutting spacing 20 mm, 30 mm and 40 mm. In the next series of tests, the cutting depth was increased to 15 mm, repeating applied cutting spacing values from the previous series. For both series, it was noticed that for the spacing t = 40 mm full spalling of the cut groove does not occur and for the depth g = 15 mm strong reactions occur leading to the construction vibrations that disturb the mining process. Mean values of component forces calculated during left and right mining of the artificial concrete block at depth g = 15 mm and at mining spacing t = 20, 30 and 40 mm are presented in Figure 16.

A smaller load on the plate with disk tools occurred during the mining from the left side [30]. The results of the laboratory tests confirm the results of the simulation tests. When the rock is attacked with the inclined surface of the mini-disk tool, located from the side of the free rock surface, almost two times lower tool load value was observed.

The last test was performed on the sandstone and granite samples. Due to the samples' relatively low volume, the cuts only from the right with a constant cutting depth were carried out. The cutting depth was set for sandstone at g = 40 mm at the mining spacing t = 30, 40, 50 and 60 mm and for granite g = 25 mm at t = 10, 20 and 30 mm.

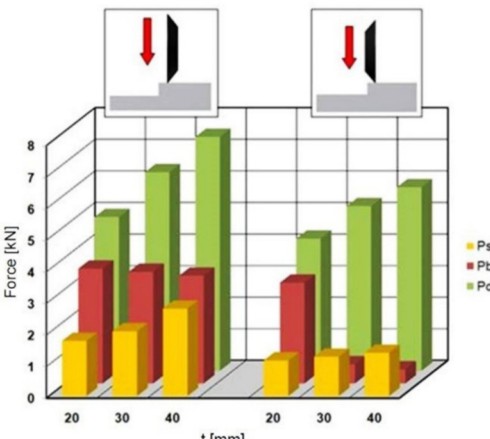

**Figure 16.** Mean values of the pressure Pd, cutting Ps and side Pb forces during mining from the left and right side of the block at mining depth g 15 mm [30].

Compared to the results of the tests for sandstone sample, it can be concluded that a significant decrease in the mining process dynamics compared to tests of concrete mining was achieved. The sandstone body's surface uncovered as a result of mining was relatively regular, which meant smooth chipping off the rock (Figure 17). In the case of granite sample mining, the pressure force Pd value was the greatest. The cutting force Ps value was comparable for all samples, while the side force Pb value achieved the highest level for sandstone sample. For both natural rock samples, the greater dimension of output grains was obtained. The obtained results clearly illustrate the influences of mining parameters on values of mining resistance component forces.

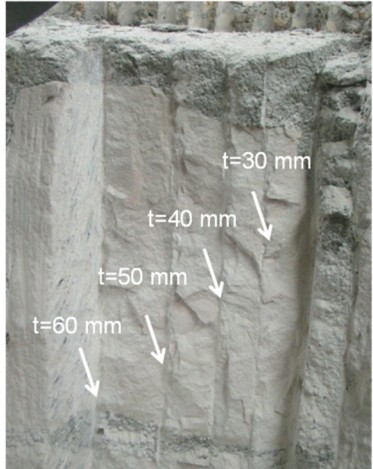

**Figure 17.** A view of the sandstone block surface after completing the mining tests in the vertical plane [30].

Presentation of mean values of mining forces for tests of mining concrete and natural samples, in which the highest values of cutting depth and spacing were obtained, is shown on a diagram in Figure 18 [30].

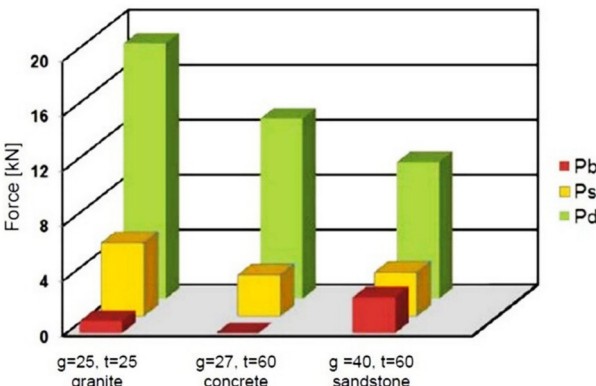

**Figure 18.** Mean values of mining forces for maximal values of mining depth and mining spacing during mining different rock samples in the vertical plane [30].

The tests' results confirm that the most convenient mode of rock mining using the plate's presented solution with the disk tools is the mining in the vertical direction when the rock is attacked with the inclined surface of the disk from the side of the free rock surface.

*5.3. Test Results on the Stand for Mining with a Complex-Motion Trajectory Milling Plate, Equipped with Asymmetrical Mini-Disk Tools*

Complementary studies were conducted to confirm this statement on the special test stand for mining tools testing (Figure 19). The test stand was adapted for assembly of the plate with disk tools and drive unit. The test stand consists from the mainframe, traverse with the support and the tool holder. The rotary table with the mined rock sample is located in the test stand axis. Instead, the tool holder, the special device for the fixing of a disk plate with drive unit was developed. It allowed the mining of the ring rock sample on the side surface with the rock sample's complex rotary movement and the plate with disk tools. The same set of disk tools was used—diameter D = 160 mm and edge angle 40° [30].

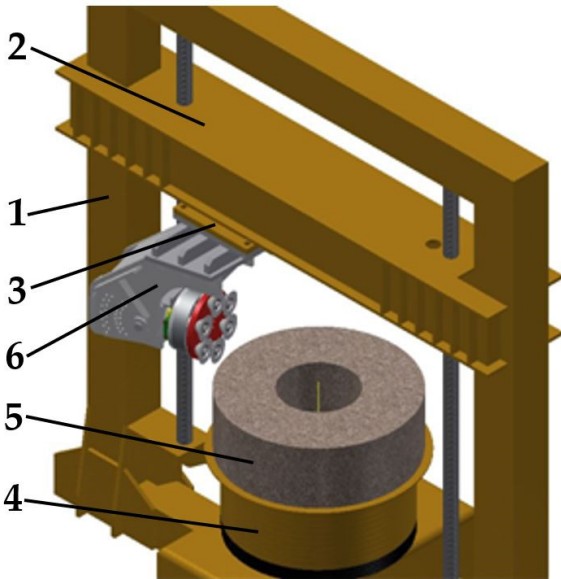

**Figure 19.** The 3D model of a lab stand for testing mining process of compact rocks with a complex-motion trajectory milling plate, equipped with asymmetrical mini-disk tools: [29]. 1—mainframe, 2—traverse, 3—support with the tool holder, 4—rotary table, 5—mined rock sample, 6—special device for the fixing of a disk plate.

Initial tests were carried out for determining the direction of revolutions of the rock sample and the plate with disk tools. Convergent and opposite directions of rotation were

checked at the plate's constant number of revolutions with disk tools at about 60 rpm and table rotation with the ring sample at about 20 rpm. The sample was mined from the top with the depth of chipping d =15 mm.

In the convergent direction of rotation, the disk tool penetrated relatively gently into the sample, breaking a fragment of concrete. In the opposite direction of the rotation, the sample and the disk tool collided quite violently. It resulted in generating significant dynamic loads and vibrations of the test stand. The convergent direction of rotation was selected for further mining trials.

Four depths of chipping were selected for further tests, d = 15, 25, 35 and 45 mm. The first mining test view with a plate with disk tools with a diameter of 160 mm and an angle of 40 ° at a chipping depth of 15 mm is shown in Figure 20a. Figure 20b shows the result of a ring sample mining test along with its entire height, with a chipping depth of 30 mm. Can be seen that during the mining, cyclic grooves appeared on the surface of the sample, related to the regularity of successive chipping of the concrete material by disk tools. There were no significant signs of wear to disk tools observed. As a result of the mining, an output with a significant granulation was obtained, depending on the depth of the chipping, with size from several to over 100 mm.

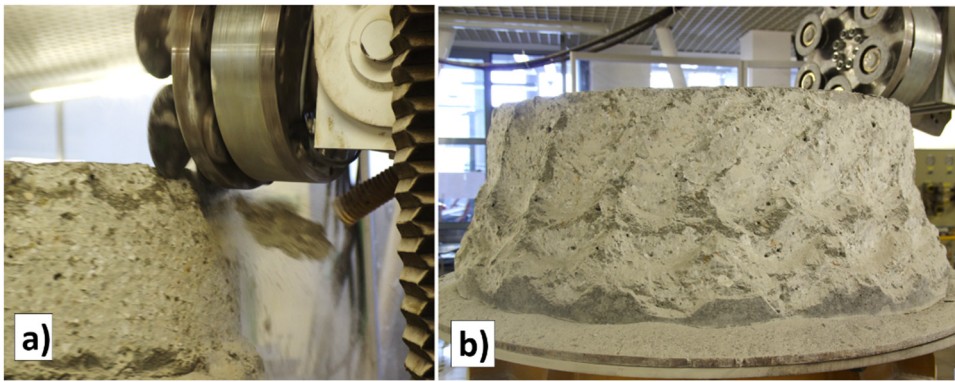

**Figure 20.** The view of the sample: (**a**) during mining at the chipping depth of 15 mm, (**b**) the surface after mining at the chipping depth of 15 mm.

The test results confirmed that usage of asymmetrical mini-disk tools allows the effective rock mining, with a large grain size of the obtained output. But the selection of mining parameters is essential. Based on the test results, the vertical direction of mining, convergent directions of rotation at a constant number of revolutions of the plate with disk tools at about 60 rpm and table rotation with the ring sample at about 20 rpm and the way of mining with the inclined surface of the disk from the side of the free rock surface, were choose.

## 6. The Use of Mini Asymmetrical Disk Tools in Roadheader Innovative Mining Head

Based on the results of computer modelling and stand tests, in the AGH Kraków Department of Machinery Engineering and Transport, the innovative solution of mining head with mini-disk tools of complex motion trajectory was developed. The mining head construction and its principle of work are presented in the scheme in Figure 21 [30,31].

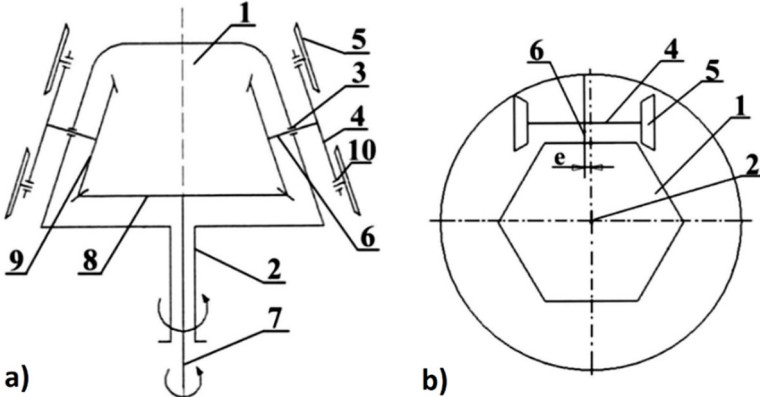

**Figure 21.** The scheme of the innovative solution of mining head with mini-disk tools of complex motion trajectory (description in the text): (**a**) kinematic diagram (**b**) top view [30].

The elaborated mining head consists of independently propelled body and mounted in its propelled plates with asymmetrical disk tools. The unit body—1 is propelled by an external drive shaft—2. In the body, in seats—3 drive shafts—6 are mounted with plates—4, on which in bearing seats—10 disk tools—5 are installed. The most favourable number of tools should be 6 to 8 pieces. The drive shafts—6 are propelled by an internal drive shaft—7 independent from the external drive shaft—2 and a set of bevel gears—8 and 9 or alternative ones. In the conception of the new solution it is crucial to ensure dislocation of the drive shafts axes—6 of the plates—4 with mining tools—5 by the value e so that they do not cross the axis of the head body—1 drive shaft—2. The plates axes' eccentric location with tools should enable their easy slotting into the rock body at the unit motion both vertical and horizontal one.

Based on the above presented solution, the new head solution with disk tools of complex trajectory was manufactured. The works were performed with cooperation with the REMAG Ltd. Company (Halifax, UK)—the leading Polish producer of light and medium roadheaders. Was planned to work out the new head solution for a medium roadheader KR150 manufactured by REMAG. The drives for both motions of mining head were separate to enable independent control of their rotational speed's value and direction. The head body was propelled by a hollow shaft using an electric motor, whereas an external shaft will realise the drive of the plates with disk tools through a gear placed inside the body, using a set of four hydraulic motors. The view of the new mining head solution, mounted on the KR150 roadheader and ready to tests, are presented in Figure 22a.

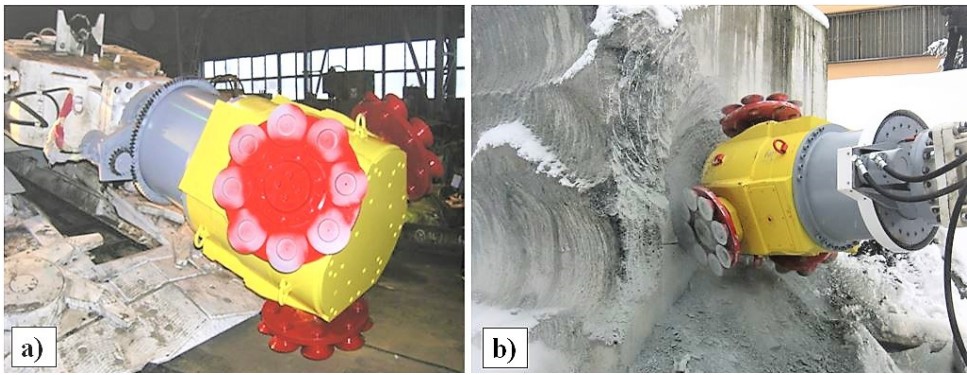

**Figure 22.** The view of the new mining head solution (**a**) mounted on the KR150 roadheader and ready to tests, (**b**) during one of the conducted mining tests [30,31].

Preliminary field tests of the head with asymmetrical disk tools of complex trajectory were conducted on a test stand prepared at the REMAG Company. The large-size concrete

block of uniaxial compressive strength of about 40 MPa was mined. The dimensions of the concrete block were 6000 mm in width, 3000 mm in depth and 4500 mm in height. Based on the earlier obtained results of the tests, for the initial tests the mining head body rotations was planned on the level about 20 rpm and for the plates with disk tools about 60–70 rpm. Disk tools were mounted on the plates with the flat side out. Mining effectiveness and the level of the tools wear were checked as well as granulation of the obtained winning. A view of the KR150 miner with the new solution during one of the conducted mining tests was shown in Figure 22b [2].

The new mining head solution worked for parameters as mentioned above without serious reservations. A view of the obtained output is shown in Figure 23a and a specific surface of the mined concrete block (intersection of the grooves) is shown in Figure 23b [2,30]. There were noticed no many symptoms of the disk tools wear.

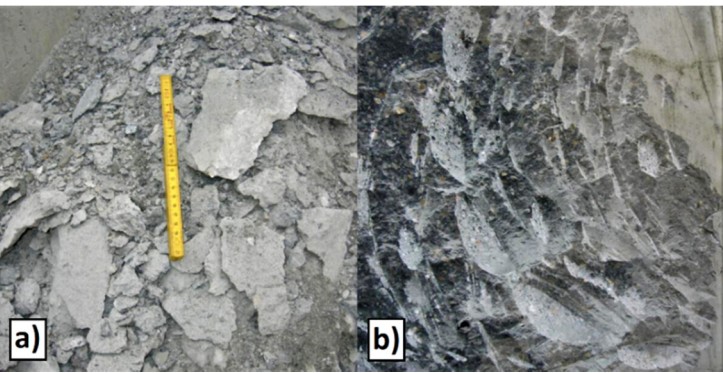

**Figure 23.** View of the mining results with the use of a new mining head: (**a**) the view of the obtained output, (**b**) the view of the specific surface of the mined concrete block (intersection of the grooves) [31].

The maximal mining web for the suggested mining head solution should not exceed 15–20 cm. For larger values of the web, temporary stops in the rotation of the head body were observed.

## 7. Analysis of the Research Results

The primary simulation research on the modelling of the mining process with asymmetrical mini-disk tools, which has been carried out with the use of Discrete Element Method (DEM) in LS-Dyna, confirmed the ability to break off the rock with reduced energy consumption (lower pressing and side force values) and obtained the output with significant grain size. The research results allowed a preliminary estimation of the mini-disk tools' geometric parameters (diameter: 160 mm and edge angle: 40°) and the cutting process parameters (mining spacing: 15 and 25 mm).

The mining process verification, carried out on laboratory stands, confirmed a good correlation of the results obtained during the simulation and laboratory tests. Observations of the process of rock destruction in simulation tests, especially the shape of the destruction zone, allow identifying many similarities to the laboratory tests results, especially during single mini-disk pressing into the rock sample. All of the tests were completed by chip separation, whose dimensions (height, depth) were many times higher than the assumed mining spacing.

For mining using the plate with mini-disk tools, the tests' results confirm that the most convenient mode of rock mining, using the plate with the disk tools, is the mining in the vertical direction when the rock is attacked with the inclined surface of the mini-disk tool, located from the side of the free rock surface. Even at the mining depth of over 20 mm and mining spacing of 40 mm, full spalling of the cut groove occurred, with almost two times lower tool load value than in the case of its reverse clamping—with the inclined surface of the mini-disk tool located from the mined sample side. The obtained results

clearly illustrate the influences of mining parameters on values of component forces of mining resistance.

The estimated and measured side force values and the kinematic parameters of the mining process (the number and direction of the rotations of the plate with disk tools and the concrete sample) were used in designing a new mining head solution. Such an approach allowed selecting the method of mounting and bearing the disk tools and selecting drive units—motor power and the type of gears and their ratio values.

The developed solution of the new mining head was positively verified during field tests. As a result of chipping, the output of considerable size was obtained. But it requires selecting the correct parameters of the mining process—the number and direction of rotation of plates with disk tools and the head body. The plates' rotation direction with disk tools and head body rotation must be opposite, and the ratio of the plates rotation and head body rotation should be approximately 3 to 1. The mining head web value is also significant. The maximal mining web for the suggested mining head solution should not exceed 15–20 cm.

## 8. Conclusions and Summary

The mechanical mining method efficiency depends mostly on the excavated rock medium, its structure and strength parameters. The use of cutting tools, mainly rotary-tangential picks, is limited to rocks with the uniaxial compressive strength of usually no more than 120 MPa, and with naturally weakened surfaces, containing no substantial inclusions of abrasive minerals. In more difficult mining conditions, the risk of sparking and dusting and the tools wear increases significantly. The impact of the excavated rock structure and the presence of abrasive materials, such as sphaerosiderites and siliceous compounds, on the cutting tool load and working life, as well as on mining efficiency, have been described in the literature [3–6,32].

For hard rock mining, symmetrical disk tools are the optimum solution. However, the huge pressure forces exerted by these tools and the size, weight, and costs of the TBM on which these tools are mounted restrict this method's application to driving of long galleries.

Based on the industrial test analysis using asymmetrical disk tools with a diameter of 400 mm, performed by the Wirth Company with so-called Hinterschneiden machine, another interesting solution of hard rock mechanical mining—undercutting or back incision method can be used. By taking advantage of the lower shear strength of the rock to break it, this method reduces energy consumption while providing per-tool efficiency that is several times greater than that of disk mining tools which work by applying static crumpling to crush the rock. It also allows a large grain size of the output.

The above-mentioned mining method benefits were the base for a development and elaboration in the AGH Kraków Department of Machinery Engineering and Transport, the mining method using asymmetrical mini-disk tools.

The results of laboratory tests presented in the described article constitute unique information regarding the impact of geometric parameters of disk tools and mining process parameters on the reaction force generated during mining. From a practical point of view, the presented test results can be input data for design purposes and can be used to select the necessary parameters of dynamic mining heads. The relations between the disk tools individual geometric parameters, the mining process and the reaction forces, identified as a result of the conducted research, can be used to validate analytical and simulation mining models.

Preliminary tests have shown great potential for increasing mining efficiency by using the mining head with mini-disk tools. However, this requires selecting kinematic parameters that will not generate excessively large reaction forces acting on the mining head and the machine body. This will enable a better control and steering of the mining process.

The conducted research significantly broadens the scope of knowledge in the field of mining with asymmetrical disk tools. The practical (utilitarian) aspect of described laboratory research work involves test results for the selection of kinematic and dynamic

parameters of a new type of the roadheader mining head equipped with disk tools. The research results on the prototype mining head confirmed the validity of the methodology used for testing with a single disk and plate with disk tools. In turn, the completed research cognitive goal was achieved by identifying the correlation between a number of disk geometrical parameters, process tools and the values of mining resistance forces.

**Author Contributions:** Conceptualization, D.P. and K.K.; methodology, K.K. and G.S.; software, G.S.; validation, G.S. and K.K.; formal analysis, G.S. and K.K.; investigation, G.S. and K.K.; resources, G.S. and K.K.; data curation, G.S. and K.K.; writing—original draft preparation, K.K. and G.S.; writing—review and editing, D.P.; visualization, G.S. and K.K.; supervision, K.K. and D.P.; project administration, D.P.; funding acquisition, D.P. All authors have read and agreed to the published version of the manuscript.

**Funding:** Model and laboratory tests carried out as part of the statutory activity titled "Selected issues of design and exploitation of machine and anthropotechnical systems in conditions of excavation, transport and processing of mineral raw materials" no. 16.16.130.942. Field tests carried out as a part of the project "Technological initiative I" no. 13444—"A new generation mining head with disk tools with a complex motion trajectory for mining compact and very compact rocks" supported by polish National Centre for Research and Development.

**Institutional Review Board Statement:** Not applicable.

**Informed Consent Statement:** Not applicable.

**Conflicts of Interest:** The authors declare no conflict of interest.

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
