# Peer review of "Study and Application of Asymmetrical Disk Tools for Hard Rock Mining"

_energies, doi:10.3390/en14071826_

Round 1

Reviewer 1 Report

The article contains very important issues concerning the mechanical mining of rocks with high compressive strength. This is an important aspect because rocks that are difficult to mining are very often exploited with the use of explosives. Moreover, the daily progress that can be increased by replacing explosives with new mining tools is important for both economic and technological reasons. Laboratory tests in several configurations of the new cutting tool deserve for special attention. Below are some comments to the article:

Line 54: “… . During rock mining, energy is used to break the rock's structure by creating cracks and craters… ..”, please write does craters represent discontinuous deformation?

Line 61: provide reference for Rittinger's low.

In the third Chapter, concerning numerical modeling with the use of the LS-Dyna, you should describe boundary conditions and failure criterion, which were adopted in the calculations.

Line 248, please write down what apparatus and instruments were used for the measurements.

Lines 244-251 and 255-262 are the same: Each of the mining tests consisted of setting an appropriate mining spacing by the appropriate aligning and locking of the handle together with the disk tool......

In the Fourth Chapter, under Figure 15, please specify the size of the concrete and rock samples.

Lines 436-437: please write whether the speed of 60 rpm and 20 rpm corresponds to the requirements in industrial conditions.

In fifth Chapter , similarly to fourth Chapter, please write what size were the samples on which the mining tests were carried out with new tools.

Author Response

Responses to the comments of Reviewer # 1.

After considering the comments of reviewers, two new chapters: No 2 "Purpose, course and methodology of the research" and No 8 “Conclusions and summary” were added to the amended paper, figures 5 and 23 were deleted, and figures 21 and 22 were merged. This affects the structure of the post-revised article. The numbers of lines, chapters and figures have been changed. The main changes in the manuscript have been marked in red.

Line 54: “… . During rock mining, energy is used to break the rock structure by creating cracks and craters… ..”, please write does craters represent discontinuous deformation?

Due to the structure of the rock medium - elastic-plastic material, cracks and craters have a discontinuous deformation.

Line 61: provide reference for Rittinger's low.

Typo error in the text - it should be law instead of low - Rittinger's law. The law that energy needed to reduce the size of a solid particle is directly proportional to the resultant increase in the surface area.

In the third chapter, concerning numerical modelling with the use of the LS-Dyna, you should describe boundary conditions and failure criterion, which were adopted in the calculations.

The boundary conditions and failure criterion used in the calculations have been described in the text in chapter No 4.

Line 248, please write down what apparatus and instruments were used for the measurements.

The text has been supplemented by a list of apparatus and instruments used for the measurements.

Lines 244-251 and 255-262 are the same: Each of the mining tests consisted of setting an appropriate mining spacing by the appropriate aligning and locking of the handle together with the disk tool......

Lines 255-262 have been delated in the text

In the fourth Chapter, under figure 15, please specify the size of the concrete and rock samples.

In the fifth Chapter, similarly to the fourth Chapter, please write what size were the samples on which the mining tests were carried out with new tools.

In chapters No 4 and 5, the dimensions of the excavated rock samples have been supplemented.

Lines 436-437: please write whether the speed of 60 rpm and 20 rpm corresponds to the requirements in industrial conditions.

The proposed mining method is a new solution in terms of kinematics - it cannot be compared with mining using standard mining heads. But the rotation speed of the roadheaders' mining head reaches 60 rpm.

Reviewer 2 Report

Dear authors

I have read the paper “The research and application of asymmetrical disk tools for hard rock mining” submitted to the Journal Energies, from the MPDI publisher. The work addresses an important issue in the mining field, related to energy optimizations by means of undercutting instead milling methods.

I have to say I am a little confused with the paper structure. The introduction chapter is correct in the first paragraphs, because it gives an interesting overview of the techniques used in this research; it starts to look weird when I cannot find the methodology chapter or the material description chapter. The Chapter 3 still explains the State of the art as if it is a Introduction chapter. Then I realize that each chapter has its own introduction, material description, methodology, results and conclusions. Chapter 4 explain goals, and again, methodology and state of the art, results and conclusions.

I think in the present form, the paper cannot be review. Introduction chapter have to gather all related to the state of the art about the research field, including objectives and goals. It is necessary a chapter with material description and the whole methodology. And, then, the results, interpretation and conclusions.  I can understand that the tests described and performed in this work has a chronological order, but it has to be presented in a scientific manner.

The figures have to be improved. The issue is not the quality, rather is the correct identification of each one: for example, figure 1 has to have A) , B) and C) , and each ones have to be described in the figure caption. Figure 3 has A) and B), but there are three figures, and no identification in the first one. All figures have to be called in the manuscript. Even if it is 3A or 3B. Figure 24 has A) and B) too. Description of the figure 24, I recommend to put a table next to the figure or below the figure. An also, this is methodology description.  

I encourage the authors to rebuild this paper, it is very interesting topic, and the results look good, but in the present format, I cannot review the work. You can find more comments in the attached edited PDF file. 

Best regards

Author Response

Responses to the comments of Reviewer # 2.

After considering the comments of reviewers, two new chapters: No 2 "Purpose, course and methodology of the research" and No 8 “Conclusions and summary” were added to the amended paper, figures 5 and 23 were deleted, and figures 21 and 22 were merged. This affects the structure of the post-revised article. The numbers of lines, chapters and figures have been changed. The main changes in the manuscript have been marked in red.

Taking into consideration the comments of the Reviewer, I would like to inform you that I made the suggested corrections in the article. 

Chapter No 2 entitled "Purpose, course and methodology of the research" was added in the Article, which describes its individual parts in sequence. 

The state of the art about the research field is presented in chapter 3.

In the last part of the article, after chapter no 7 “Analysis of research results” chapter No 8. “Conclusions and summary” is added. 

The figures mentioned in the review have been corrected. Descriptions a), b) etc. have been introduced in figures 1, 3, 8, 16, 24, 26. 

Descriptions of elements in figures 13 and 20 have been placed under the figures and their numbering in the text has been removed.

In figure 7 - The capital letter A indicates the enlargement of detail A - Disk tool edge.

Figure 9 and 12 have been corrected - the description was increased and axes improved. Figure 9 presents our own research results.

However, I cannot fully agree with the comments that Chapter 3 still explains the state of the art as it is an introduction chapter. Chapter 4 explain goals, and again, methodology and state of the art, results and conclusions. In chapter No 3 (currently No 4) the modelling of the rock mining process with the use of asymmetrical disk tools was described. In chapter No 4 (currently No 5) the results of verification tests of the rock mining process with the use of mini asymmetrical disk tools at the laboratory stands are presented.

Reviewer 3 Report

I have reviewed an article the title “The research and application of asymmetrical disk tools for hard rock mining”. The topic of the manuscript falls within the scope of the journal. The manuscript is well written, concise, scope, and objectives are clearly defined. The manuscript in its present form needs some minor revision. The following are my comments

  1. Describe results in abstract. How modeling was performed? Describe in abstract
  2. Too much self-citations. Reduce number of self-citations
  3. English language needs massive improvement
  4. On line 188 Tensile strength Rr=2,3? Please check the numbers
  5. Too many figures. Reduce number of figures and better way is to add text with proper reference
  6. A conclusion and summary section needs to be added
  7. Figure 9: The word force is written twice
  8. Describe the steps involved in this work with a complete workflow diagram.

Author Response

Responses to the comments of Reviewer # 3.

After considering the comments of reviewers, two new chapters: No 2 "Purpose, course and methodology of the research" and No 8 “Conclusions and summary” were added to the amended paper, figures 5 and 23 were deleted, and figures 21 and 22 were merged. This affects the structure of the post-revised article. The numbers of lines, chapters and figures have been changed. The main changes in the manuscript have been marked in red.

  1. Describe results in abstract. How modelling was performed? Describe in abstract

In the Chapter concerning numerical modelling with the use of the LS-Dyna short description about the way of modelling has been supplemented.

  1. Too much self-citations. Reduce number of self-citations

Two self-citations - item number 22 and 32, quoted in the text, have been deleted.

  1. English language needs massive improvement

In response to the reviewer's note regarding the improvement of the English language, I would like to inform you that the text of the article has been grammatically checked using the professional Grammarly program and has been checked in terms of the vocabulary used in the text by a person professionally associated with the mining industry and having a certificate of knowledge of the English language.

  1. On line 188 Tensile strength Rr=2.3? Please check the numbers

The values of uniaxial compressive strength (UCS) Rc = 24 MPa and tensile strength (BTS) Rr = 2,3 MPa are correct.

  1. Too many figures. Reduce number of figures and better way is to add text with proper reference

Figures No 5 and 23 have been deleted. Figures 21 and 22 have been combined into one figure.

  1. A conclusion and summary section needs to be added

In the last part of the article, after chapter No 7 “Analysis of research results” chapter No 8. “Conclusions and summary” has been added.

  1. Figure 9: The word force is written twice

It should be side force instead of force force.  

  1. Describe the steps involved in this work with a complete workflow diagram.

Chapter No 2 entitled "Purpose, course and methodology of the research activities" was added in the Article, which describes its individual parts in sequence.

Round 2

Reviewer 2 Report

Dear authors

The paper “The research and application of asymmetrical disk tools for hard rock mining” has been reviewed for a second time. This study is pretty interesting, and can contributing to the development of new techniques for rock fragmentation.  

I have just a few comments, listed below:

Line 78-79: Unnecesary phrase

Line 88: please change verb conjugation

Figure 3 : I think there are two figures here, thus I recommend to differenciate them as a) and b)

Line 145: Mg is tonnes ?

Figure 4 : I also see three pics here. a) b) and c)

Lines 189-190: Please, re-build this phrase, is not easy to understand

Ficure : Sorry, but I can also see three figures in this pics, a), b) and c)

Line 202: In which article? this article ?

Figure 8: I understand that in this point the collapse of the  rock is produced (during simulation) would be interesting to remark this milestone in the graphic.

Line 275:  a space is missing

Line 282-283: What does this symbol mean? division ?  ÷

Figure 11: What's the link between simulation and laboratory test? I'm thinking about comparatives among virtual and real test, using forces results as a shared results when your imputs are similar: for example, same disk tool geometry and same spacing. 

Line 372: Type error , should be 14 and after a point

Figure 16: Are these results comparable with simulations?

563: type error, extra-brackets !!!

You can also find these comments in the attached PDF. 

Best regards 

Author Response

Dear Sir,

Thank You very much for the review. In the new, attached version of the article, I made the suggested corrections. I have described them one by one below.

Line 78-79: Unnecesary phrase

Phrase has been delated

Line 88: please change verb conjugation

The verb conjugation has been changed as follows:

The Discrete Element Method (DEM) with the computer package LS-Dyna was used to carry out these studies.

Figure 3 : I think there are two figures here, thus I recommend to differenciate them as a) and b)

I would leave the designations a, b and c, as in the descriptions of the other drawings.

Line 145: Mg is tonnes ?

According to the SI system of measures, the weight is given in Mg (equal to a ton)

Figure 4 : I also see three pics here. a) b) and c)

The figures mentioned in the review have been corrected. Descriptions a), b) etc. have been introduced in figures 4 and 5.

Lines 189-190: Please, re-build this phrase, is not easy to understand

The phrase has been changed as follows:

Most studies were carried out during the simulation of rock mining with the use of conical picks. 

Line 202: In which article? this article ?

Yes In this article

Figure 8: I understand that in this point the collapse of the  rock is produced (during simulation) would be interesting to remark this milestone in the graphic.

The comment has been added in the text as follows:

A sudden drop in the value of the forces in the graphs means the moment when the rock fragment was chipped from the block

Line 275:  a space is missing

Space has been added.

Line 282-283: What does this symbol mean? division ?  ÷

This symbol ÷ denotes the measuring range: from 0 to 200 kN, from 0 to 300 mm.

Figure 11: What's the link between simulation and laboratory test? I'm thinking about comparatives among virtual and real test, using forces results as a shared results when your imputs are similar: for example, same disk tool geometry and same spacing.

The comment has been added in the text as follows:

Comparing the results obtained during simulation tests and laboratory tests, for the same mining parameters, e.g. disk diameter, mining spacing, etc., it can be stated that they are very compatible. The difference in the value of the forces does not exceed 15%.

Line 372: Type error , should be 14 and after a point

The error has been corrected.

Figure 16: Are these results comparable with simulations?

The comment has been added in the text as follows:

The results of the laboratory tests confirm the results of the simulation tests. When the rock is attacked with the inclined surface of the mini-disk tool, located from the side of the free rock surface, almost two times lower tool load value was observed.

563: type error, extra-brackets !!!

The error has been corrected.

Thank You

Best Regards

Authors